# Circadian control of hepatitis B virus replication

Xiaodong Zhuang [1,12✉], Donall Forde[1,12], Senko Tsukuda[1,2,3], Valentina D'Arienzo[1], Laurent Mailly [4], James M. Harris [1], Peter A. C. Wing [1], Helene Borrmann [1], Mirjam Schilling [1], Andrea Magri[1], Claudia Orbegozo Rubio[1], Robert J. Maidstone[5,6], Mudassar Iqbal[7], Miguel Garzon[8], Rosalba Minisini [9], Mario Pirisi[9], Sam Butterworth[8], Peter Balfe [1], David W. Ray [5,6], Koichi Watashi[2,3,10], Thomas F. Baumert [4,11] & Jane A. McKeating [1✉]

Chronic hepatitis B virus (HBV) infection is a major cause of liver disease and cancer worldwide for which there are no curative therapies. The major challenge in curing infection is eradicating or silencing the covalent closed circular DNA (cccDNA) form of the viral genome. The circadian factors BMAL1/CLOCK and REV-ERB are master regulators of the liver transcriptome and yet their role in HBV replication is unknown. We establish a circadian cycling liver cell-model and demonstrate that REV-ERB directly regulates NTCP-dependent hepatitis B and delta virus particle entry. Importantly, we show that pharmacological activation of REV-ERB inhibits HBV infection in vitro and in human liver chimeric mice. We uncover a role for BMAL1 to bind HBV genomes and increase viral promoter activity. Pharmacological inhibition of BMAL1 through REV-ERB ligands reduces pre-genomic RNA and de novo particle secretion. The presence of conserved E-box motifs among members of the Hepadnaviridae family highlight an evolutionarily conserved role for BMAL1 in regulating this family of small DNA viruses.

[1] Nuffield Department of Medicine, University of Oxford, Oxford, UK. [2] Department of Virology II, National Institute of Infectious Diseases, Tokyo, Japan. [3] RIKEN Cluster for Pioneering Research, Wako, Japan. [4] University of Strasbourg and Inserm, UMR-S1110, Institut de Recherche sur les Maladies Virales et Hépatiques, Strasbourg, France. [5] NIHR Oxford Biomedical Research Centre, John Radcliffe Hospital, Oxford, UK. [6] Oxford Centre for Diabetes, Endocrinology and Metabolism, University of Oxford, Oxford, UK. [7] Division of Informatics, Imaging and Data Sciences, Faculty of Biology, Medicine and Health, University of Manchester, Manchester, UK. [8] Division of Pharmacy and Optometry, School of Health Sciences and Manchester Academic Health Sciences Centre, University of Manchester, Manchester, UK. [9] Department of Translational Medicine, Università del Piemonte Orientale, Novara, Italy. [10] Department of Applied Biological Sciences, Tokyo University of Science Graduate School of Science and Technology, Japan and Institute for Frontier Life and Medical Sciences, Kyoto University, Kyoto, Japan. [11] Pôle Hépato-Digestif, Institut Hopitalo-Universitaire (IHU), Hopitaux Universitaire de Strasbourg, Strasbourg and Institut Universitaire de France, Paris, France. [12] These authors contributed equally: Xiaodong Zhuang, Donall Forde. ✉email: xiaodong.zhuang@ndm.ox.ac.uk; jane.mckeating@ndm.ox.ac.uk

Hepatitis B virus (HBV) is a global health problem, with more than 250 million people chronically infected and at least 880,000 deaths/year from HBV-related liver diseases such as cirrhosis and hepatocellular carcinoma (WHO, Global hepatitis Report 2017). Current treatments only suppress viral replication and are not curative due to the persistence of viral DNA in hepatocytes. Chronic infection is associated with a blunted host innate immune response[1] and exhausted anti-viral T cell responses that fail to control HBV replication[2,3]. In most cases, treatment is life-long and patients may still develop liver cancer[4]. Effective anti-viral drugs have revolutionized treatments for hepatitis C virus and there is a growing impetus to identify similar curative therapies for HBV[5].

Viral entry into a host cell represents the first step in the infectious life cycle and the discovery that sodium taurocholate co-transporting polypeptide (NTCP) is a receptor for HBV[6,7] enabled the development of culture systems that support viral infection and therapeutic options. Synthetic peptides mimicking the viral encoded glycoprotein preS1 binding site for NTCP, such as Myrcludex-B (MyrB) inhibit HBV infection[8,9]. A recent phase II clinical trial showed its efficacy in HBV patients co-infected with hepatitis delta virus (HDV)[10]. Following NTCP-dependent entry into hepatocytes, the encapsidated partially double-stranded relaxed circular DNA (rcDNA) genome is repaired into cccDNA via host DNA repair pathways[11]. This episomal DNA persists in the nucleus as a long-lived nucleosome-associated minichromosome[12] and its transcriptional activity is essential in defining HBV replication in the liver. HBV gene expression is dependent on RNA polymerase II complex and a repertoire of activators and repressors reported to regulate transcription (reviewed in ref. [13]), however, our understanding of the role environmental factors play in regulating HBV replication is not well understood.

The circadian clock is a ubiquitous endogenous timing system that coordinates physiological processes that define daily rhythms of metabolism and inflammation[14]. The circadian signaling pathway in mammals exists in nearly every cell and is primarily controlled by a series of transcription/translation feedback loops. The transcription activators BMAL1 (brain and muscle ARNT-like 1) and CLOCK (circadian locomotor output cycles kaput) drive thousands of transcripts including their own repressors. The nuclear hormone transcriptional repressors REV-ERBα and REV-ERBβ are controlled by BMAL1 and provide a negative feedback loop to control gene expression in a tissue-dependent manner[15] (Fig. 1a). As obligate intracellular parasites viral replication is reliant on the host cell machinery and recent reports highlight a pleiotropic role for BMAL1 in promoting human immunodeficiency virus[16,17] and hepatitis C virus[18] replication whilst suppressing herpes, influenza[19] and respiratory syncytial viruses[20] (reviewed in ref. [21]). These findings highlight a dynamic role for circadian pathways in defining host susceptibility to viral infection, however, the molecular mechanisms are not understood.

HBV replicates in the liver where approximately 20% of hepatic genes show a rhythmic expression pattern[22], suggesting that the virus has successfully evolved to persist in this circadian entrained organ. Cortisol, a circadian regulated glucocorticoid hormone is associated with HBV reactivation in immunosuppressive settings[23,24]. Lipid and bile acid metabolism are circadian regulated including the expression of transcription factors, peroxisome proliferator-activated receptor and sterol regulatory element-binding protein (reviewed in ref. [25]) that have been reported to activate HBV transcription[26–28]. The liver-specific transcription factor HNF4α has been linked to HBV tropism[29] and interacts with circadian components BMAL1/CLOCK[30] and REV-ERB[31]. These studies highlight the importance of understanding the role of circadian pathways in regulating HBV replication and persistence.

In this study we demonstrate a direct role for REV-ERB to repress NTCP expression and HBV uptake into naïve target cells. Furthermore, we show that pharmacological activation of REV-ERB inhibits HBV replication in vitro and in vivo. We demonstrate that BMAL1 binds HBV DNA and regulates viral genome transcription and genesis of viral particles. Our studies provide innovative routes to better understanding and therapeutic targeting in chronic HBV infection.

## Results

**REV-ERBα regulates sodium taurocholate co-transporting polypeptide expression.** To investigate a role for circadian pathways in the HBV life cycle we first assessed whether NTCP expression is rhythmic. We selected the bipotent HepaRG cell line that can be differentiated to biliary-like epithelial cells as well as hepatocyte-like cells that is frequently used for drug toxicity studies[32,33], expresses NTCP[34] and supports HBV replication[35]. Different methods can be used to induce circadian gene expression and to reset the phase of the cellular circadian clock ex vivo[36]. We compared two commonly used approaches, dexamethasone and serum shock, to synchronize HepaRG cells expressing a Bmal1 promoter-luciferase construct. Both agents induced a rhythmic cycling of Bmal1 promoter that was reduced by treatment with the synthetic REV-ERB agonist SR9009[37] (Supplementary Fig. 1). Furthermore, we demonstrate that SR9009 had no cytotoxic effect on HepaRG cells in the dose range tested (Supplementary Fig. 2). Synchronized differentiated HepaRG cells (dHepaRG) show a circadian cycling of Bmal1/Rev-Erbα transcripts (Fig. 1b) and NTCP showed a rhythmic pattern of expression in phase with BMAL1 (Fig. 1c). Since REV-ERB is the major transcriptional repressor of BMAL1, we evaluated a role for REV-ERB in regulating NTCP and showed that silencing Rev-Erbα increased NTCP mRNA levels (Fig. 1d). Consistent with the known transcriptional feed-back loops between the circadian activators and repressors we show that silencing Bmal1 increased NTCP mRNA (Supplementary Fig. 3). Overexpressing Rev-Erbα in HepaRG cells significantly reduced NTCP mRNA levels (Fig. 1e), supporting a role for REV-ERBα as a transcriptional repressor of NTCP.

To assess whether REV-ERBα regulates the NTCP promoter we co-transfected HepaRG cells (Fig. 2a) and human Huh-7.5 cells (Supplementary Fig. 4) with an NTCP promoter-luciferase construct (−1143 to +108 of the human NTCP gene[38]) and a Rev-Erbα expression plasmid and showed a significant reduction in promoter activity (Fig. 2a). Activating REV-ERB with SR9009 reduced NTCP mRNA and protein expression and impaired bile salt transport in dHepaRG cells (Fig. 2b, c). In contrast, SR9009 treatment had no effect on epidermal growth factor receptor expression (Supplementary Fig. 5) a host factor recently identified to modulate NTCP-mediated HBV internalization[39]. REV-ERB and BMAL1 regulate gene expression by binding ROR response elements (RORE) or E-boxes, respectively, in the promoter and enhancer regions of their target genes[40,41]. The NTCP promoter region contains two RORE motifs (Fig. 2d) and we assessed whether REV-ERBα binds the promoter by chromatin immunoprecipitation and quantitative PCR (ChIP-qPCR). We show REV-ERBα binding above the control IgG with a significant enrichment in RORE1 motif within the NTCP promoter and its host target Bmal1 promoter (Fig. 2d). We noted five putative E-box motifs in the NTCP promoter (Fig. 2e), however, ChIP-qPCR experiments failed to show any evidence for a direct interaction of BMAL1 with the NTCP promoter. In summary, our data identifies a direct role for REV-ERBα as a repressor of NTCP expression and function.

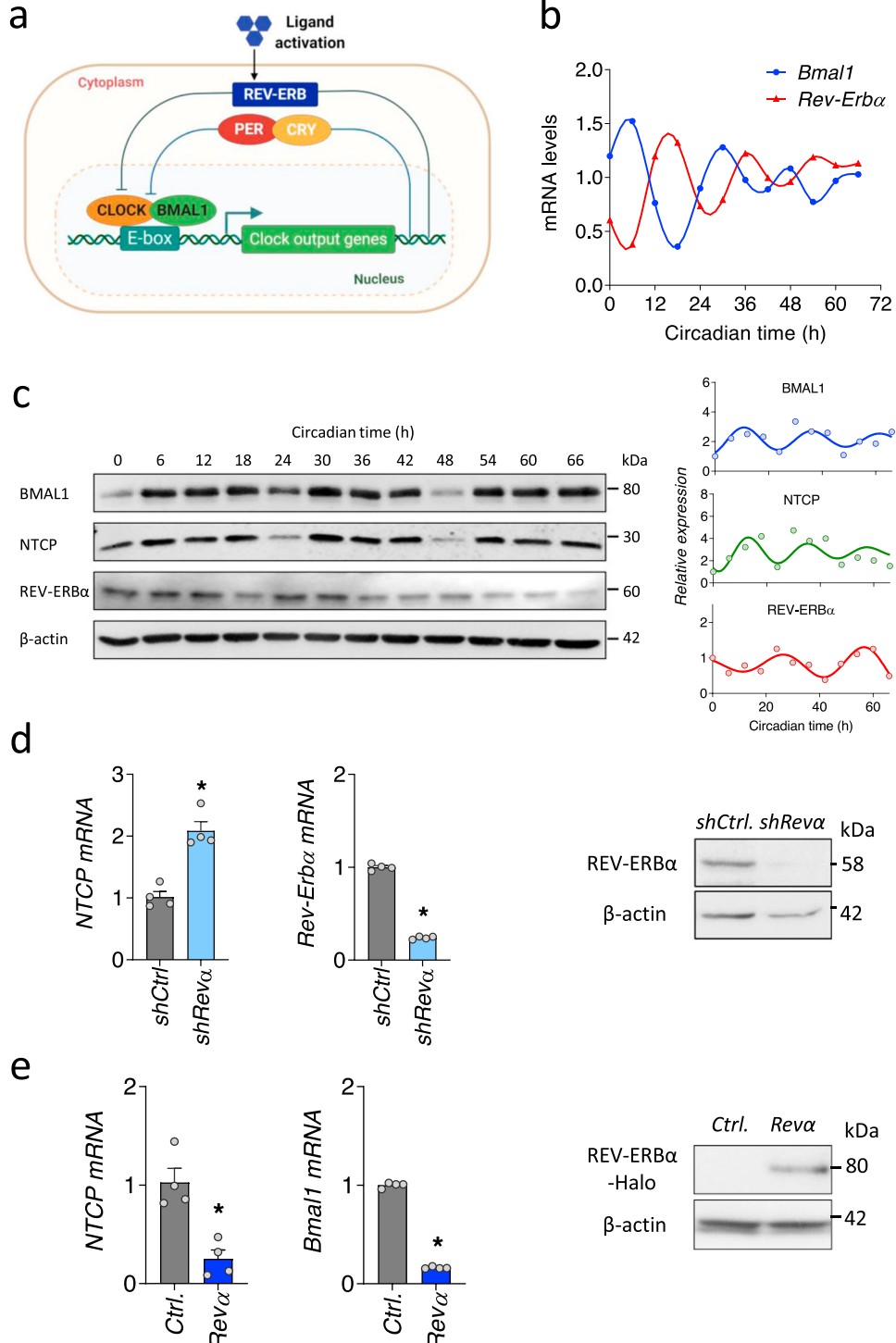

**Fig. 1 REV-ERBα coordinates the circadian expression of NTCP. a** Circadian transcription–translation feedback loops generate daily rhythmic expression of circadian genes. The molecular clock generates a cycle of 24-hour periodicity when a heterodimeric transcription factor BMAL1-CLOCK activates the transcription from E-box sequences in target gene promoters. Gene products feedback to repress the transcriptional activity of the heterodimeric activator. Ligand activation of REV-ERB represses the transcription of Bmal1. **b** Synchronization of differentiated HepaRG cells (dHepaRG). dHepaRG cells were treated with dexamethasone at 100 nM for 2 h and samples collected at 6 h intervals. Bmal1 and Rev-Erbα mRNAs were measured by quantitative reverse transcription polymerase chain reaction (qRT-PCR) and expressed relative to the mean. Data are the average of two independent experiments. **c** Synchronized dHepaRG cells were assessed for BMAL1, NTCP and REV-ERBα expression together with housekeeping β-actin by western blotting. Densitometric analysis quantified BMAL1, NTCP and REV-ERBα in individual samples and was normalized to their own β-actin loading controls. Data are the average of two independent experiments. **d** Total RNA was extracted from control or Rev-Erbα silenced HepaRG cells and NTCP and Rev-Erbα mRNA levels measured by qRT-PCR. REV-ERBα expression together with housekeeping gene β-actin were assessed by western blotting. Data are expressed relative to control (mean ± SEM, $n = 4$, Mann–Whitney test, Two-sided). **e** Total RNA was extracted from control or Halo-tagged Rev-Erbα overexpressed HepaRG cells and Bmal1 and NTCP mRNA levels measured by qRT-PCR. Data are expressed relative to control (mean ± SEM, $n = 4$, Mann–Whitney test, Two-sided). *$p < 0.05$. Data are provided in the accompanying Source Data file.

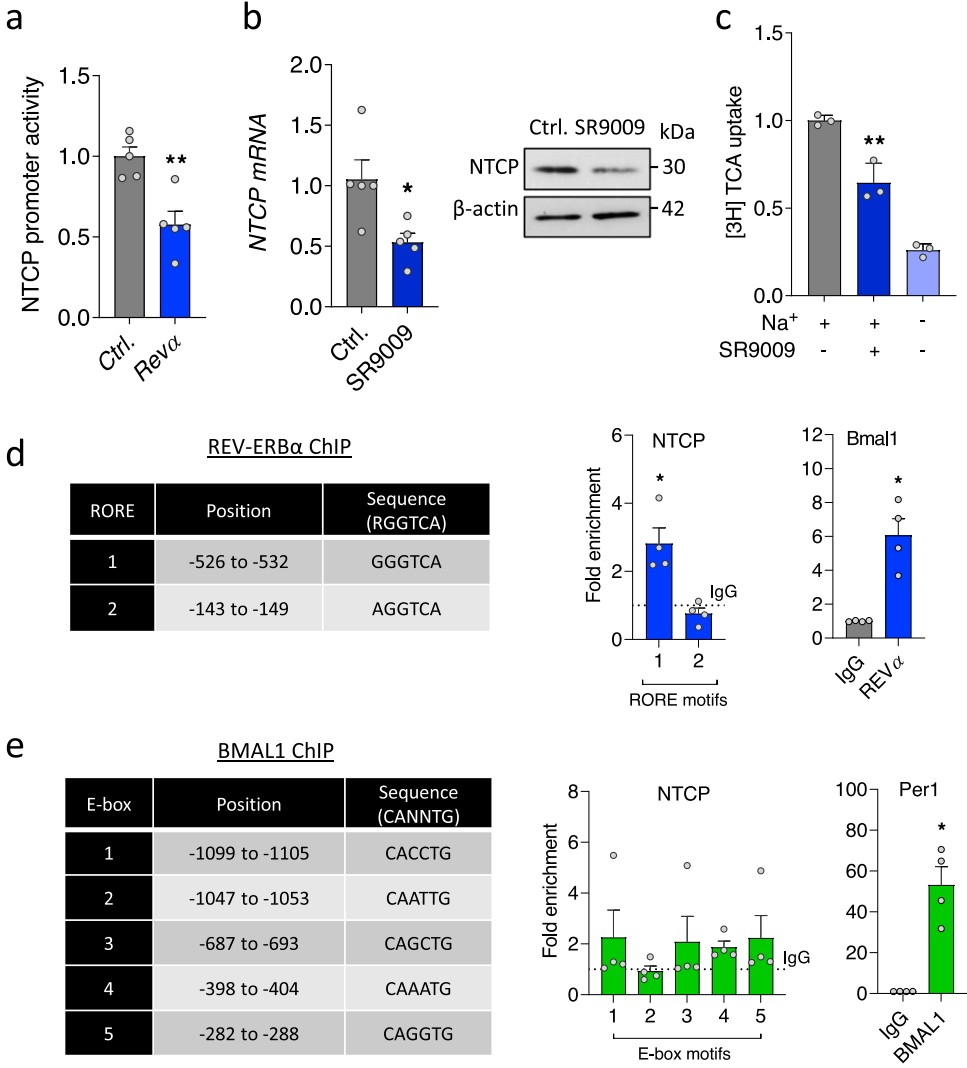

**Fig. 2 Direct role for REV-ERBα to bind and regulate NTCP. a** HepaRG cells were co-transfected with NTCP promoter luciferase reporter (−1kb) and control or Halo-tagged *Rev-Erbα* expression plasmid. NTCP promoter activity was determined 48 h later by quantifying luciferase activity and data expressed relative to control (mean ± SEM, *n* = 5, Mann–Whitney test, Two-sided). **b** dHepaRG cells were treated with SR9009 (20 μM) for 24 h and *NTCP* RNA and protein measured along with the housekeeping β-actin by qRT-PCR or western blotting (mean ± SEM, *n* = 5, Mann–Whitney test, Two-sided). **c** dHepaRG cells were treated with SR9009 (20 μM) for 24 h; [3H]-taurocholic acid was supplied with media for 10 min; intracellular radioactivity was measured by liquid scintillation counting and data expressed relative to control (mean ± SEM, *n* = 3, One-way ANOVA with multiple comparisons, Two-sided). Coordinates of RORE or E-box motifs in NTCP promoter. Chromatin extracts from HepaRG cells were immunoprecipitated using antibodies specific for REV-ERBα (**d**), BMAL1 (**e**) or rabbit IgG as a negative control. PCR for NTCP promoter regions using primers targeting defined RORE or E-box motifs and control host genes *Bmal1* or *Per1* was performed and %IP data presented relative to a rabbit IgG control shown as the dotted line (mean ± SEM, *n* = 4, Mann–Whitney test, Two-sided). \**p* < 0.05, \*\**p* < 0.01. Data are provided in the accompanying Source Data file.

**Pharmacological activation of REV-ERB inhibits hepatitis B and Delta virus infection**. To assess whether REV-ERB regulated NTCP expression impacts HBV infection, firstly we quantified cell surface NTCP expression using fluorescent MyrB and show that SR90009 treatment significantly reduced receptor expression (Fig. 3a). Using a recently developed synchronized infection protocol[42] we show that SR9009 treatment reduced HBV uptake into dHepaRG cells (Fig. 3b). To evaluate whether these observations can be recapitulated in a full de novo HBV infection system, we pre-treated dHepaRG cells with SR9009 followed by HBV infection and measured viral replication by PCR quantification of intracellular RNAs and viral antigen expression. The basal core promoter (BCP) drives transcription of pre-core and pre-genomic RNAs from two start sites that are only 70 base pairs apart[43] and since our PCR cannot discriminate between these viral-encoded RNAs, we label transcripts as pC/pgRNA to

represent the sum of both RNAs. SR9009 treatment of naïve dHepaRG cells significantly reduced HBV cccDNA, pC/pgRNA and viral encoded antigens (HBeAg and HBsAg) (Fig. 3c). Hepatitis Delta virus (HDV) also utilizes NTCP to enter hepatocytes, providing a model system to independently validate our results. As expected, treating dHepaRG cells with SR9009 inhibited HDV infection as measured by antigen expression and viral RNA (Fig. 3d). In summary, these data show that activating REV-ERB with the synthetic ligand SR9009 inhibits both HBV and HDV infection of naïve target cells.

**REV-ERB agonists inhibit HBV transcription**. HBV infected hepatocytes are long-lived in the chronic infected liver[12] and we were interested to investigate a role for REV-ERB agonists to regulate HBV DNA replication. SR9009 treatment of HBV infected dHepaRG cells resulted in a significant reduction in pC/

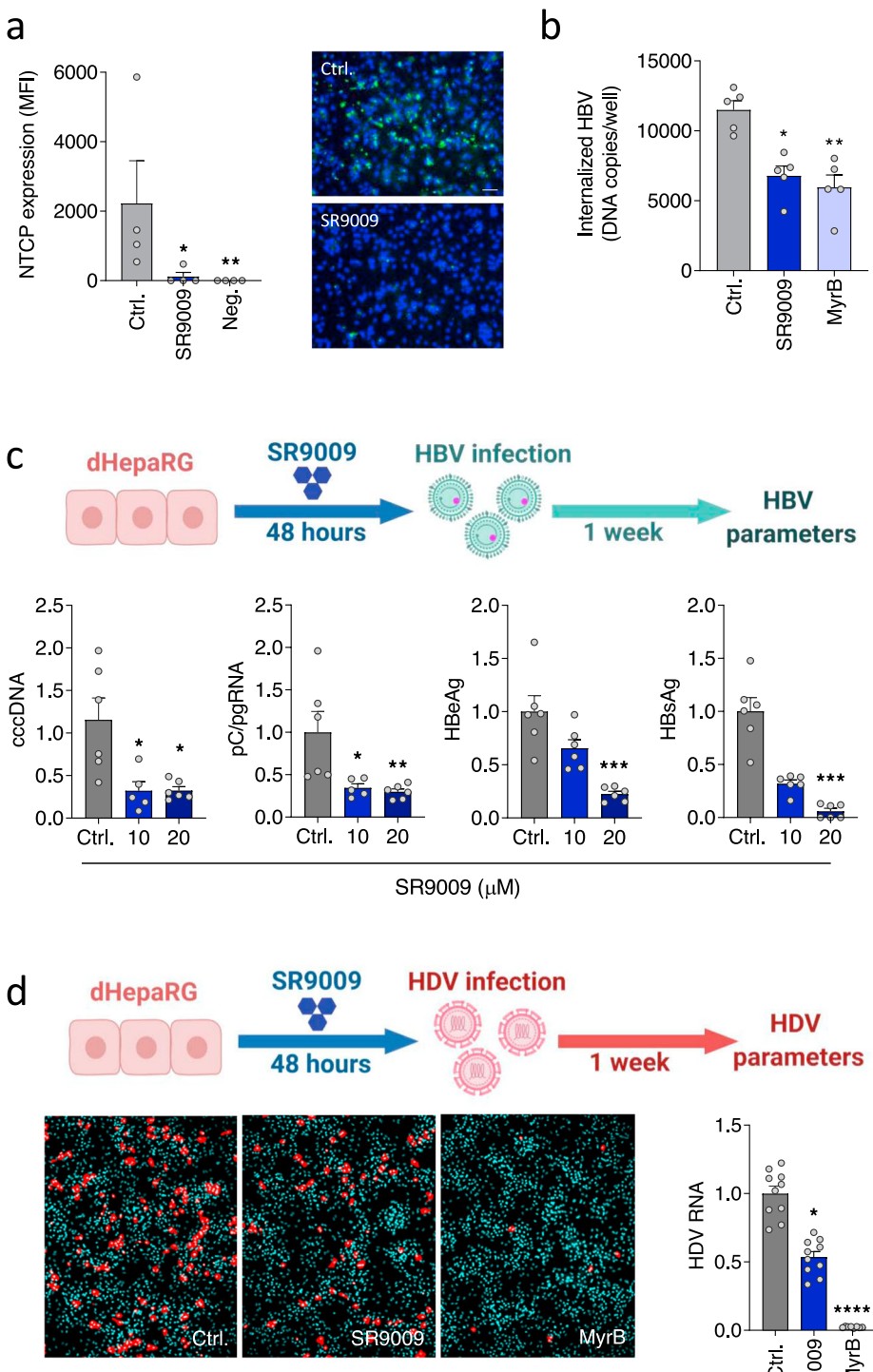

**Fig. 3 Pharmacological activation of REV-ERB inhibits HBV and HDV infection. a** dHepaRG cells were treated with SR9009 (20 μM, 48 h) and NTCP expression assessed using Myrcludex B (MyrB) tagged with Alexa 488 fluorophore. Unstained cells were used as the negative control. Median fluorescence intensity was quantified (mean ± SEM, $n = 4$, Kruskal–Wallis ANOVA with Dunn's test). Representative fluorescent images are shown (scale bar = 100 μm). **b** dHepaRG cells were treated with SR9009 (20 μM) or MyrB (200 nM) and inoculated with HBV at 4 °C for 3 h. Unbound virus was removed by washing and the inoculated cells incubated at 37 °C for 16 h, trypsinized to remove non-internalized virus and intracellular HBV DNA levels quantified by qPCR and presented as copies per well (mean ± SEM, $n = 5$, Kruskal–Wallis ANOVA with Dunn's test). **c** dHepaRG cells were treated with SR9009 (10 or 20 μM) for 48 h and inoculated with HBV for 6 h. 1 week post-infection, cccDNA, pC/pgRNA, HBeAg and HBsAg were quantified by qRT-PCR or ELISA assay. In all cases, data are expressed relative to untreated (Ctrl) cells (mean ± SEM, $n = 6$, Kruskal–Wallis ANOVA with Dunn's test). **d** dHepaRG cells were treated with SR9009 (20 μM) for 48 h and inoculated with HDV and 7 days post-infection, HDV antigen (HDAg) and viral RNA measured by fluorescent staining or qRT-PCR. MyrB was included as the positive control to assess NTCP-dependent HDV infection (mean ± SEM, $n = 8$–10, Kruskal–Wallis ANOVA with Dunn's test). $*p < 0.05$, $**p < 0.01$, $***p < 0.001$. Data are provided in the accompanying Source Data file.

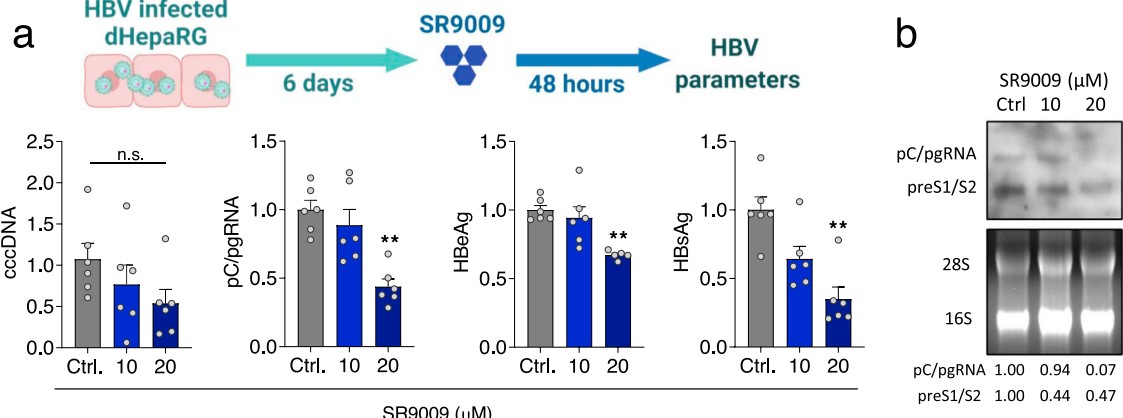

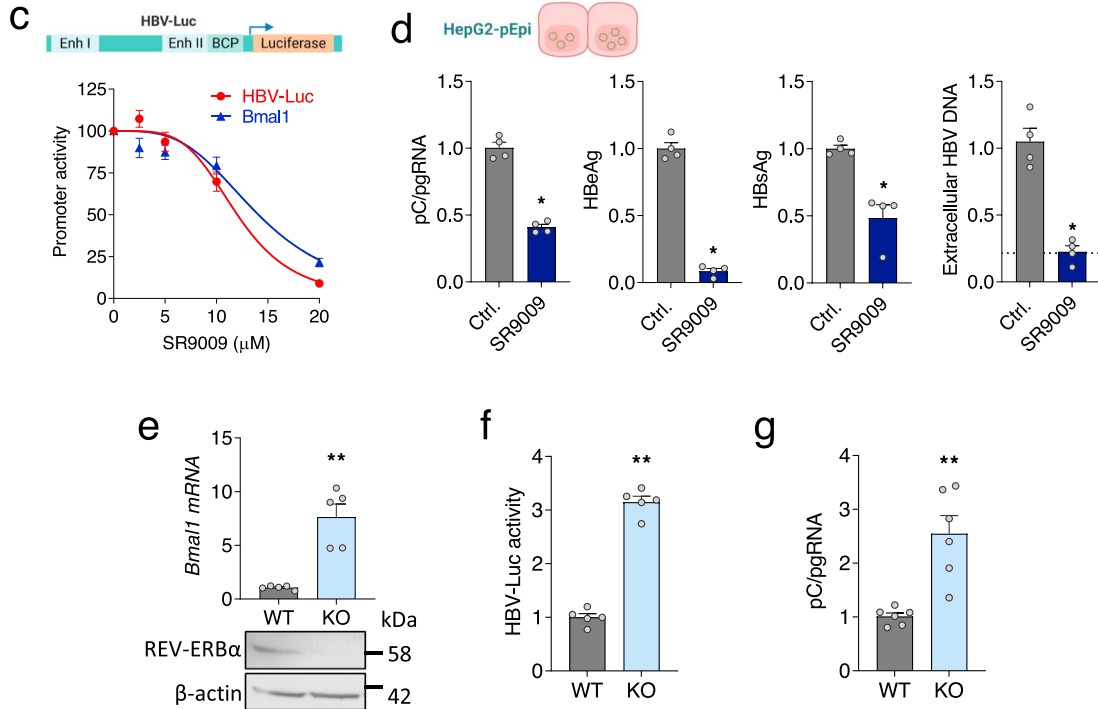

**Fig. 4 REV-ERBα regulates HBV transcription. a** dHepaRG cells were infected with HBV for 6 days followed by SR9009 treatment (10 or 20 μM) for 48 h. HBV parameters: cccDNA, pC/gRNA, HBeAg and HBsAg were determined by qRT-PCR or ELISA assays. In all cases, data are expressed relative to untreated (Ctrl) cells (mean ± SEM, $n = 6$, Kruskal–Wallis ANOVA with Dunn's test). **b** HBV transcripts in the SR9009 treated dHepaRG cells were quantified by northern blotting. Densitometric analysis quantified pC/pgRNA and preS1/S2 transcripts and was normalized to the 28 S and 18 S rRNA loading controls. **c** HepG2-NTCP cells were transfected with HBV EnhI/II-BCP or Bmal1 promoter luciferase reporters for 48 h and treated with SR9009 for 24 h and promoter activity quantified by measuring luciferase activity (mean ± SEM, $n = 4$–6). **d** HepG2-pEpi cells were treated with SR9009 (20 μM) for 48 h. HBV parameters: pC/pgRNA, HBsAg, HBeAg and extracellular particle HBV DNA were determined by qRT-PCR or ELISA assays. The reverse transcriptase inhibitor Entecavir (ETV, 1 μM) was included as positive control for the genesis of de novo particles and is represented by the dashed line. In all cases, data are expressed relative to untreated (Ctrl) cells (mean ± SEM, $n = 4$, Mann–Whitney test, Two-sided). **e** Parental (WT) or *Rev-erbα* KO HepG2-NTCP cell lysates were assessed for REV-ERBα expression together with housekeeping gene β-actin by western blotting. Total RNA was extracted and *Bmal1* mRNA levels measured by qRT-PCR. Data are expressed relative to WT cells (mean ± SEM, $n = 5$, Mann–Whitney test, Two-sided). **f** WT or *Rev-erbα* KO cells were co-transfected with HBV BCP luciferase reporter and Renilla luciferase reporter and 48 h post transfection the cells lysed and luciferase activity quantified. Data are expressed relative to WT (mean ± SEM, $n = 5$, Mann–Whitney test, Two-sided) and the data normalized for transfection efficiency. **g** WT or *Rev-erbα* KO HepG2-NTCP cells were infected with HBV and pC/pgRNA measured by qRT-PCR 72 h later. Data are expressed relative to WT (mean ± SEM, $n = 6$, Mann–Whitney test, Two-sided). *$p < 0.05$, **$p < 0.01$. Data are provided in the accompanying Source Data file.

pgRNA, HBeAg and HBsAg but no change in cccDNA levels (Fig. 4a), suggesting a role for the agonist to repress viral transcription. The HBV genome is transcribed from four promoters (BCP, Sp1, Sp2 and Xp)[13] that drive six major RNAs: pC that

encodes HBeAg; pgRNA that is translated to yield core protein and polymerase; preS1, preS2 and S RNAs encoding the surface envelope glycoproteins and X transcript for the multi-functional x protein. Northern blotting showed that SR9009 treatment

 ARTICLE

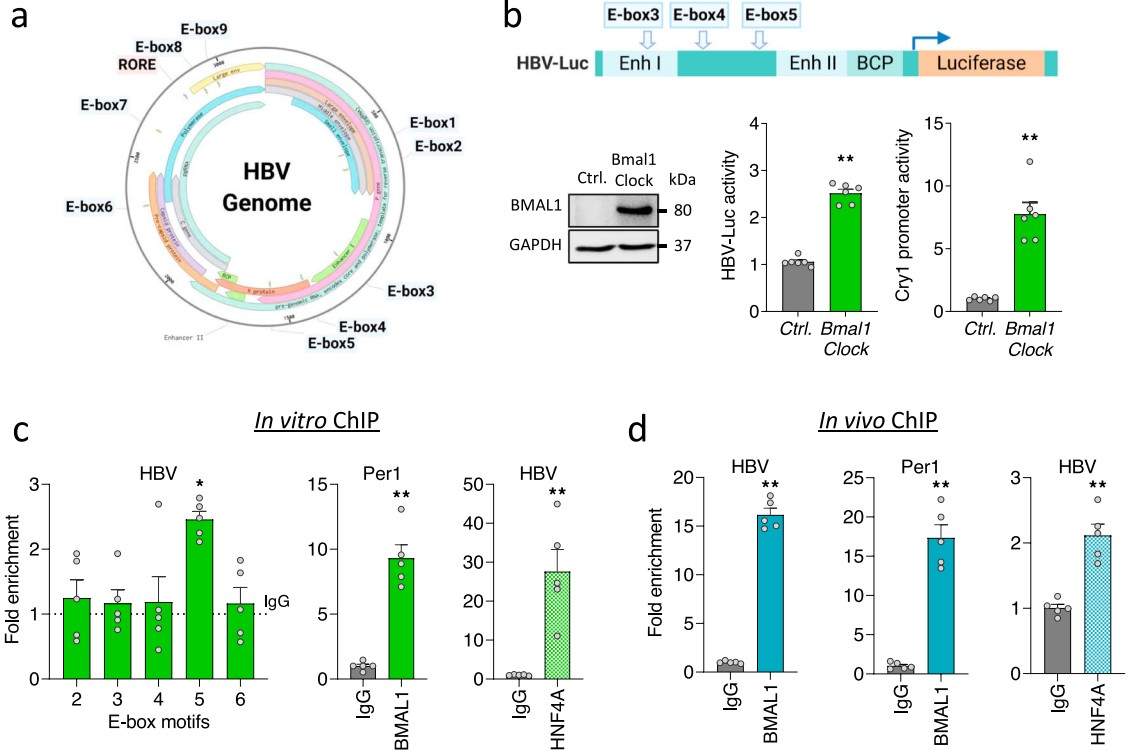

**Fig. 5 BMAL1 regulates HBV transcription by direct binding to the viral genome. a** E-box and RORE motifs in the HBV genome (Genotype D, Ayw,
NC_003977.2). **b** HepG2-NTCP cells were co-transfected with HBV EnhI/II-BCP or Cry1 promoter luciferase reporters and *Bmal1*-Flag/*Clock* expression
plasmids. 48 h later cells were lysed and promoter activity determined by quantifying luciferase activity. Cell lysates were probed with anti-Flag to assess
exogenous BMAL1 expression together with anti-GAPDH by western blotting. Data are expressed relative to control treatment (mean ± SEM, $n = 6$,
Mann–Whitney test, Two-sided). **c** Chromatin extracts from HepG2-pEpi cells were immunoprecipitated using antibodies specific for BMAL1, HNF4A or
rabbit IgG as a negative control. PCR for HBV DNA using primers targeting defined E-Box motifs or control host gene Per1 were performed and IP data
presented relative to a rabbit IgG control shown as the dotted line (mean ± SEM, $n = 5$, Kruskal–Wallis ANOVA with Dunn's test). **d** Chromatin isolated
from five liver samples from two HBV infected FRG-NOD mice was used for a ChIP assay with anti-BMAL1 or anti-HNF4A antibodies, along with rabbit IgG
as a negative control. PCR for HBV DNA using primers targeting E-Box 5 or control host gene Per1. %IP was calculated and data expressed relative to the
irrelevant IgG control (mean ± SEM, $n = 5$, Mann–Whitney test, Two-sided). *$p < 0.05$, **$p < 0.01$. Data are provided in the accompanying Source Data file.

reduced both pC/pg and preS1/S2 transcripts in infected dHe-
paRG cells (Fig. 4b). To extend these results and to investigate the
mechanism by which SR9009 treatment reduces viral RNAs we
transfected cells with a HBV promoter-luciferase construct
encoding firefly luciferase under the control of EnhI/II and the
BCP[44] (HBV-Luc). SR9009 treatment reduced HBV promoter
activity (Fig. 4c), demonstrating a role for the agonist to repress
viral transcription.

To further validate a role for the REV-ERB agonist in
regulating HBV transcription we used HepG2 cells harboring
episomal copies of replicating HBV genomes (HepG2-pEpi)[45]
and showed that SR9009 treatment significantly reduced pC/
pgRNA, HBeAg and HBsAg expression (Fig. 4d). Since pgRNA is
reverse transcribed to generate new encapsidated DNA genomes,
we hypothesized that the SR9009-dependent reduction in pgRNA
levels would limit the genesis of new virus particles. Treating
HepG2 pEpi (Fig. 4d) or HepG2.2.15 cells[46] (that carry integrated
copies of the viral genome) significantly reduced secreted HBV
DNA (Supplementary Fig. 6). To extend these results we
genetically knocked-out (KO) *Rev-Erbα* and assessed cellular
permissivity to support HBV transcription. HepaRG cells are
recalcitrant to CRISPR gene editing, possibly reflecting their
bipotent nature, we therefore chose to KO *Rev-Erbα* in human
hepatocyte derived HepG2 cells that were engineered to express
NTCP from an exogenous promoter[47] (Supplementary Fig. 7). As
expected, *Rev-Erbα* KO HepG2 cells showed increased levels of

*Bmal1* mRNA compared to parental or wild-type cells (Fig. 4e)
and increased HBV promoter activity (Fig. 4f). Finally, de novo
HBV infection of the *Rev-Erbα* KO cells showed a significant
increase in pC/gRNA levels compared to parental wild-type cells
(Fig. 4g). Collectively, these data show a direct role for REV-ERB
in repressing HBV promoter activity, associated with reduced pC/
pgRNA levels and particle genesis.

**BMAL1 binds HBV genome in vitro and in vivo.** To define the
mechanism underlying our earlier observation that REV-ERB
repressed HBV transcription we inspected the viral DNA genome
(Genotype D, ayw) and identified nine E-box and one RORE
motif, highlighting potential sites for interaction (Fig. 5a, Sup-
plementary Fig. 8). Three E-box motifs (E-box3/4/5) reside within
HBV-Luc promoter construct and overexpressing *Bmal1* and
*Clock* activated both the viral and endogenous host Cry1 pro-
moter (Fig. 5b). To evaluate whether BMAL1 regulates viral
transcription via these E-boxes, we isolated chromatin from the
HepG2-pEpi line for ChIP-qPCR using a recently published
protocol that can shear episomal HBV DNA, allowing provisional
mapping of binding sites[48]. Primers mapping over E-boxes 2 and
6 that are located outside the enhancer or promoter regions were
included as negative controls for the ChIP-qPCR. We observed an
enrichment of BMAL1 binding above the IgG control to E-box 5
and to the promoter region of its known host target Per1 (Fig. 5c).

Next we investigated whether BMAL1 binds HBV DNA in vivo using liver samples isolated from HBV infected human liver chimeric mice[49,50] and showed an enrichment of BMAL1 binding to HBV DNA (Fig. 5d). In contrast, we failed to demonstrate REV-ERBα binding to HBV DNA (Supplementary Fig. 9). As a control for the in vitro and in vivo ChIP, we show the well characterized liver specific transcription factor HNF4α binds HBV DNA in these samples (Fig. 5c, d). In summary, our data shows clear evidence of BMAL1 binding to HBV DNA in vitro and in vivo.

**Pharmacological activation of REV-ERB modulates the human hepatocellular transcriptome and inhibits HBV transcription in vivo.** Humanized mice generated by grafting human hepatocytes to Fah−/−, Rag2−/−, IL2Rg−/−, non-obese diabetic (FRG-NOD) immunodeficient mice have contributed to our understanding of viral hepatitis[49–51]. This mouse model provides a unique opportunity to define human and mouse transcriptional responses to REV-ERB activation. In this study, humanized FRG-NOD mice were infected with HBV for eight weeks followed by vehicle or SR9009 treatment for two weeks. At the end of the experiment, mouse liver and serum were collected to determine host responses and viral parameters. SR9009 treatment impacts gene expression in both human and mouse cells (Fig. 6a). Since Bmal1 is a primary target of the REV-ERB paralogues, the inhibition of Bmal1 transcripts in both human and mouse liver cells supports SR9009 engagement of REV-ERB in vivo (Fig. 6b). This conclusion is strengthened by finding a common direction of change in the majority of human and mouse clock genes (Fig. 6b). Analyzing differentially expressed gene ontology showed that genes regulated in both human and mouse hepatocytes were enriched for pathways involved in energy metabolism, including amino acid, lipid and mitochondrial function (oxidative phosphorylation) (Fig. 6c). These biological pathways were previously reported to show circadian variation[25], supporting the regulation of the liver clock by SR9009. In the human hepatocytes unique pathways were seen including insulin, Ras, and hypoxia signaling, that may reflect species differences or the consequence of HBV infection. Comparing the genes regulated by SR9009 in the mouse or human cells with genes affected by liver-specific loss of the REV-ERB paralogues[52] revealed a limited overlap (Fig. 6d). This most likely reflects the differences between circuits solely reliant on liver REV-ERB paralogues compared to those which are dependent on system-wide changes driven by the SR9009 acting on REV-ERBs throughout the mouse. We noted an increased overlap in genes co-regulated by SR9009 and genetic REV-ERB KO in mouse liver compared to genes in human hepatocytes (Fig. 6d), most likely reflecting shared species origin.

During the course of treatment, animal weights and human albumin were monitored and limited adverse effects were observed in both groups. Consistent with previous reports, a modest weight loss was observed in the SR9009 treated group[37,53] (Supplementary Fig. 10). Importantly, we observed a reduction in hepatic pC/pgRNA levels and peripheral HBeAg in the SR9009 treated mice (Fig. 6e). In contrast, there was limited change in HBV DNA or HBsAg levels in the SR9009 treated group (Fig. 6f) which may reflect their differential half-life in vivo. Analyzing the RNA-seq data from SR9009 and vehicle treated chimeric mice showed a minimal perturbation of known host activators or repressors reported to regulate HBV replication[54] (Supplementary Fig. 11). In summary, we show that SR9009 regulates BMAL1 and other core clock gene transcripts in human and mouse hepatocytes and significantly reduces HBV pC/pgRNA, consistent with our model showing that BMAL1 binds HBV DNA and activates viral transcription.

**Discussion**
Circadian networks shape the liver transcriptome and our work highlights a dual role for the circadian clock in HBV replication: firstly, direct evidence for REV-ERB to bind and regulate NTCP expression and pharmacological activation of REV-ERB blocks HBV/HDV entry into naïve cells; secondly, BMAL1 binds HBV DNA and increases viral promoter activity. Pharmacological inhibition of BMAL1 through REV-ERB ligands reduced pregenomic RNA and de novo particle secretion. (Fig. 7). Our ChIP experiments show BMAL1 binding to episomal copies of HBV DNA in HepG2 cells and in hepatocytes of HBV infected mice with humanized livers. We noted a greater enrichment of BMAL1 binding to HBV genomes isolated from the infected mice compared with results from HepG2 cells, suggesting that non-synchronized culture systems may be underscoring the role BMAL1 plays in regulating viral transcription. In contrast, we noted reduced HNF4α association with HBV DNA isolated from the infected mice compared to HepG2 cells. HBV encodes multiple E-boxes in its compact genome and importantly, all of the E-boxes (3-5) within the EnhI/II-BCP region are conserved among all HBV genotypes and higher primates, suggesting a conserved evolutionary role for BMAL1 to regulate these viruses. It is tempting to speculate that HBV along with other members of the *Hepadnaviridae* family have evolved to replicate and to exploit the circadian-signaling pathway to persist in the liver.

Edgar et al. reported a more severe infection of herpes simplex virus 1 and Influenza A in *Bmal1* knock-out mice[19], suggesting an anti-viral role for BMAL1. A recent study from Sengupta and colleagues[55] reported a time of day dependency in Influenza replication in mice that was predominantly mediated via *Bmal1* regulation of inflammatory pathways in viral clearance, with a minimal role for directly regulating viral replication in this acute infection model. Since chronic HBV infection is associated with an exhausted anti-viral T cell response that may have limited capacity to control viral replication[1–3], we predict a more significant role for circadian pathways in modulating HBV replication.

In the past decade, several compounds capable of activating REV-ERB were reported[56] and compounds such as SR9009 and SR9011 were shown to improve metabolic endpoints in diet-induced obese mice[37,57]. Our studies show that SR9009 regulated core clock genes in human and mouse cells in the chimeric liver mouse model. A recent report demonstrating REV-ERB dependent and independent effects of SR9009[58] suggests some additional off-target effects. We cannot exclude the possibility of additional pathways contributing to SR9009 anti-viral activity; however, our use of genetic targeting approaches unequivocally confirms a role for REV-ERB and BMAL1 in regulating HBV replication.

Our data showing rhythmic expression of the BMAL1 promoter, transcripts and protein in synchronized HepaRG cells provides a useful human liver model for in vitro circadian studies. dHepaRG cells exhibit many characteristics of human hepatocytes including the expression of key metabolic enzymes, innate immune components and drug transporters[59] and have been used to study non-alcoholic fatty liver disease[60]. NTCP showed a rhythmic expression in dHepaRG cells consistent with previous studies reporting a circadian expression in mouse liver[61,62]. Our observation that pharmacological activation of REV-ERB inhibited NTCP expression, bile acid and HBV/HDV uptake, suggests a strategy for administering antiviral drugs in a way that maximizes benefits and minimizes adverse effects[63].

Recent studies show that perturbation of circadian clock components associates with inflammatory and metabolic diseases[25,64,65]. However, the relationship between circadian-controlled gene expression and development of chronic hepatitis B (CHB) associated liver disease is poorly understood. Yang et al. reported a perturbation of core clock genes in HBV-associated

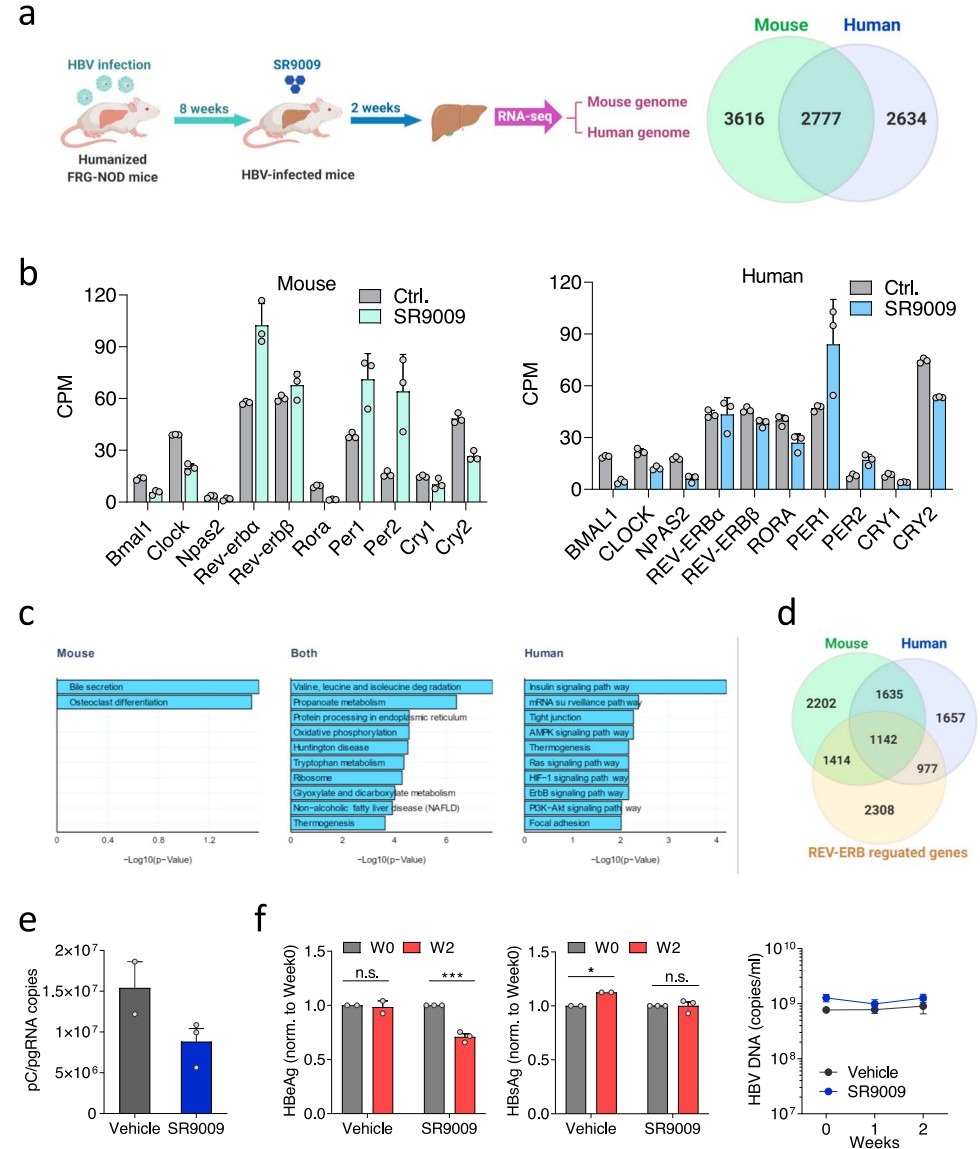

**Fig. 6 Pharmacological activation of REV-ERB modulates the human transcriptome and inhibits HBV BCP transcription in vivo. a** Fah−/−, Rag2−/−, IL2Rg−/−, non-obese diabetic (FRG-NOD) mice transplanted with primary human hepatocytes were infected with $10^9$ HBV genome equivalents purified from concentrated supernatant of the HepG2.2.15 cell line (HBV genotype D subtype ayw). Eight weeks later mice were treated twice a day by intraperitoneal injection of either vehicle or 100 mg/kg SR9009 for 2 weeks. At the end of experiment, mouse livers were collected for RNA-seq analysis and sequence reads mapped to the human or mouse genome. Differential expressed genes (DE) in human and mouse cells (19 of the DE genes in mouse had no human ortholog match). **b** Differentially expressed core clock genes in mouse and human. The counts per million (CPM) of the key clock genes assigned to the two genomes in the control and treated group are shown (mean ± SD, $n = 3$ independent liver samples). **c** Human KEGG pathway analysis of differentially expressed genes in mouse only, both mouse and human, and human only sets. Negative log10 adjusted $p$-values are given and top 10 significant (adjusted $p$-value <0.05) pathways shown. **d** Overlap of differentially expressed genes from SR9009 treated mouse livers with REV-ERB regulated genes. **e** Total RNA from mouse livers were extracted and HBV pC/pgRNA levels measured by qRT-PCR. Data are expressed relative to vehicle treated group (mean ± SEM, $n = 2$ for vehicle group and 3 for SR9009 group). **f** Peripheral HBeAg, HBsAg and HBV DNA levels were quantified (mean ± SEM, $n = 2$ for vehicle group and 3 for SR9009 group, Two way ANOVA analysis). *$p < 0.05$, **$p < 0.01$, ***$p < 0.001$. Data are provided in the accompanying Source Data file.

HCC and suggested this was mediated via the viral encoded regulatory HBx protein[66]. To ascertain if there was any perturbation of clock gene expression in patients diagnosed with CHB in the absence of HCC, we quantified transcript levels of *Bmal1* and *Rev-Erb* along with core clock genes in liver biopsies from CHB patients (Supplementary Fig. 12). We noted a modest reduction in *Bmal1* and increased *Rev-Erbα* and *Rev-Erbβ* transcripts in the HBV infected patients compared to healthy subjects, suggesting

perturbation of clock gene expression in this small cohort of patients, meriting further investigation in a larger cohort.

Our results show the key circadian transcriptional activator BMAL1 binds HBV genomic DNA and increases viral promoter activity, supporting a trans-activating role for BMAL1 in driving HBV replication. Our analysis of the human transcriptome from SR9009 treated chimeric mice showed a minimal perturbation of known host activator or repressor genes reported to regulate HBV

transcription, lending further support to this conclusion. To the best of our knowledge this is the first report of BMAL1 binding a viral genome, however, an earlier report showing that BMAL1/CLOCK over-expression activated the HIV promoter activity[17], is consistent with integrated copies of HIV binding BMAL1. HBV does not require integration into the host genome to replicate, yet, integrated viral DNA fragments are common in CHB and may contribute to carcinogenesis. Integrated copies of HBV which carry multiple circadian regulatory elements will introduce additional clock motifs to the infected cell, that may result in unwanted oscillation of certain genes or disrupted host circadian rhythms, a well-known risk factor for cancer development[67–69]. Targeting of circadian clock pathways that inhibit HBV entry and silence cccDNA transcription provide a previously undiscovered antiviral approach that complements current standard-of-care approaches and agents in development.

## Methods

**Cell lines and viruses.** The human hepatoma cell line HepG2-NTCP and HepG2-pEpi were maintained in Dulbecco's modified Eagle's medium (DMEM)/10% fetal bovine serum (FBS)/1% nonessential amino acids/penicillin/streptomycin (Invitrogen, Carlsbad, CA). HepaRG cells (Gift from Ulrike Protzer, Technische Universität München, Munich) were differentiated as previously described[70,71]. In brief, HepaRG cells were seeded in 12-well plates and maintained for 2 weeks in growth medium. Cells were cultured for an additional 2 weeks in medium that was supplemented with 1.8% v/v DMSO (Sigma–Aldrich). Differentiated HepaRG cells were synchronized by treating with 100 nM dexamethasone for 2 h. REV-ERBα KO HepG2 clones were generated by transfecting a pool of three REV-ERBα CRISPR/Cas9 KO plasmids (Santa Cruz Biotechnology, UK) followed by fluorescence activated cell sorting (FACs) and clonal expansion. Individual clones were screened for BMAL1 expression by western blotting. Purified HBV was produced from HepAD38 cells as previously reported[47].

**Reagents and antibodies.** All tissue culture media and supplements, including fetal calf serum, were obtained from Invitrogen. All tissue culture plasticware were purchased from Sarstedt. Dexamethasone (Sigma, UK) and REV-ERB agonist SR9009 (Calbiochem, US) were dissolved in dimethyl sulfoxide (DMSO) and their cytotoxicity determined by a Lactate dehydrogenase (LDH) assay (Promega, UK). The BMAL1 promoter luciferase reporter vector and REV-ERBα expression plasmid were purchased from Promega, UK and Origene, UK respectively. The NTCP promoter luciferase reporter vector was purchased from Genecopoeia, US. The Lenti-shRev-Erbα construct was a gift from Dr. B. Grimaldi, University of Genoa, Italy. The Cry1 promoter construct and Bmal1/Clock expression plasmids were a gift of Ximing Qin, Anhui University, Hefei, China. The Cry1 promoter was amplified from genomic DNA using forward primer: 5′-ATCCTCGAGGTAAAGATGCA CATGTGGCCCTG-3′ and reverse primer: 5′-CTAAAGCTTCGTCCGGAGGA CACGCATACC-3′ and cloned into the pGL3 luciferase reporter vector (Promega, UK). The Bmal1 expression plasmid was previously described[72] and further engineered with a Flag tag. The Clock expression plasmid was previously described[73]. Myrcludex B was the kind gift of Stephan Urban. The complete list of primers is provided in Supplementary Table 1, while the antibodies used are provided in Table 1.

**NTCP bile acid transporter assay.** dHepaRG cells were incubated with [$^3$H]-taurocholic acid (0.2 μM) in the presence or absence of the indicated compounds in a sodium-containing (5 mM KCl, 1.1 mM KH$_2$PO$_4$, 1 mM MgCl$_2$, 1.8 mM CaCl$_2$, 10 mM D-glucose, 10 mM HEPES, 136 mM NaCl, 20 μM taurocholate, pH 7.4] or a sodium-free buffer (the above buffer replacing 136 mM NaCl with 136 mM N-Methyl-D(-)-glutamine) at 37 °C for 10 min to allow substrate uptake into cells. The cells were washed with cold PBS and lysed with 0.05% SDS and intracellular radioactivity measured by liquid scintillation counting.

**HDV infection.** HDV was recovered from culture supernatants of Huh-7 cells transfected with pSVLD3 (kindly provided by John Taylor at the Fox Chase Cancer Center) and pT7HB2.7[74,75]. The HDV supernatant was concentrated (x50) using an Amicon centrifugal filter (MWCO: 100 kDa, Merck, Germany) dHepaRG cells were infected with HDV at a MOI of 25 genome equivalents per cell in the presence of 5% polyethylene glycol (PEG) 8000 for 24 h. The cells were washed three times with medium to remove viral inoculum and cultured for an additional 6 days. HDV infection was evaluated by detecting HDAg by immunofluorescence assay and by quantifying HDV RNA by qRT-PCR as previously described[76].

**HBV internalization.** dHepaRG cells treated with the indicated compounds were treated with HBV at a multiplicity of infection of 200 at 4 °C for 3 h to allow viral attachment to the cell surface without endocytosis in the presence of 4% PEG 8000. The cells were washed three times with PBS and cultured at 37 °C for 16 h to allow viral internalization. The cells were then trypsinized to remove cell-bound but non-internalized virus and washed three times with PBS to extract intracellular DNA for HBV quantification.

**HBV de novo infection.** HepG2-NTCP cells were seeded on collagen-coated plasticware and treated with 2.5% DMSO for 3 d before infection. The cells were incubated with HBV at a MOI of 200 genome equivalents per cell in the presence of 4% PEG 8,000 for 24 h. Viral inoculum was removed at 24 h post-infection, the cells were washed three times with PBS and maintained in DMEM 10% FCS supplemented with 2.5% DMSO. Secreted HBeAg and HBsAg were quantified by ELISA (Autobio, China).

**PCR quantification of HBV DNA and RNA.** Total cellular DNA and RNA were extracted from HBV-infected cells using the All-Prep-DNA and RNA kit (QIAGEN). qPCR of cccDNA was performed as previously described[47]. Briefly,

### Table 1 Antibodies.

| Name | Supplier | Cat. |
|---|---|---|
| Anti-BMAL1 | Abcam | Ab93806 |
| Anti-REV-ERBα | Thermo Fisher | PA5-29865 |
| Anti-β-actin | Sigma | A5441 |
| Anti-NTCP | ATLAS | HPA042727 |
| Anti-HDAg | Scrum | n/a |
| Anti-HaloTag | Promega | G9211 |
| Anti-BMAL1 (ChIP grade) | Abcam | Ab3350 |
| Anti-HNF4A (ChIP grade) | Abcam | Ab181604 |
| Rabbit IgG | Sigma | NI01 |

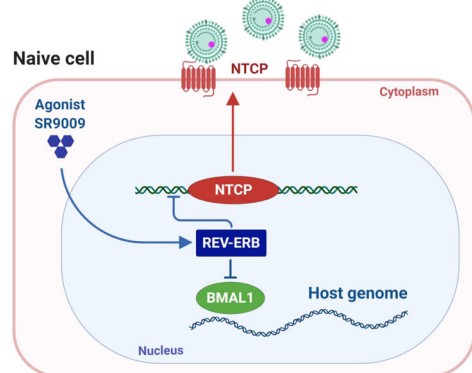
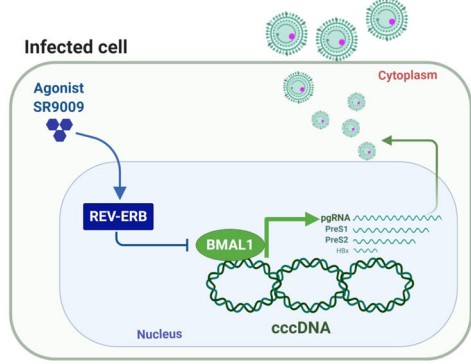

**Fig. 7 Model of circadian clock component REV-ERB and BMAL1 regulating HBV life cycle.** The circadian repressor REV-ERB regulates HBV and HDV entry into hepatocytes through modulating viral receptor NTCP expression. BMAL1 binds the HBV genome and promotes transcription and particle genesis. Activating REV-ERB with synthetic agonists or protein overexpression inhibits BMAL1 and represses HBV replication.

extracted DNA was treated with 5U of T5 exonuclease (NEB) at 37 °C for 30 min followed by heat inactivation at 95 °C. Treated DNA samples were amplified in a SYBR green qPCR reaction (PCR Biosystems). For HBV DNA quantification, a 10-fold dilution series of a HBV plasmid was used. HBV pC/pgRNA was quantified from DNase-treated RNA extracts using a one-step reverse transcriptase qPCR (RT-qPCR) kit (Takyon). HBV primer probes (FAM) were used to amplify pC/pgRNA, with primer probes for β 2 microglobulin (VIC) used as an internal control in a multiplexed RT-qPCR reaction. All qPCR reactions were carried out on a Roche LightCycler 96 (Roche).

**Northern blotting**. Samples were analyzed as described previously[47]. Briefly, RNA from de novo infected dHepaRG cells was extracted using Trizol Reagent (Life Technologies) and 10 μg of purified RNA electrophoresed in a 1% MOPS agarose gel containing 2.2 M formaldehyde. 18 S and 28 S ribosomal RNA species were visualized under UV light after electrophoresis to verify the amount of RNA loaded and to assess degradation. After denaturation (50 mM NaOH for 5 min). RNAs were transferred to a nylon membrane by capillary transfer using 20× SSC buffer. Membranes were washed and RNAs fixed by UV crosslinking. To detect HBV RNA, membranes were hybridized at 65 °C overnight with a digoxigenin-labeled DNA probe covering the entire HBV genome and visualized using a luminescent DIG detection kit (Roche).

**HBV promoter assay**. HepG2-NTCP cells were transfected using Fugene-6 (Promega) with promoter constructs driving luciferase (pGL3-HBV-Luc) and after 48 h promoter activity determined by quantifying luciferase expression (Promega).

**ChIP and quantitative PCR**. $1 \times 10^7$ HepG2-pEPI cells were harvested from 80% confluent 15 cm plates, or mouse liver tissue samples thawed and homogenized. Cells or homogenized tissue samples were fixed with 1% formaldehyde (Sigma–Aldrich 47608) for 10 min at room temperature before quenching with 125 mM glycine. Cells were washed twice with ice cold PBS, pelleted (150 g, 10 min 4 °C) and lysed in 500 μl of Nuclear Extraction buffer (10 mM Tris-HCl (pH 8.0), 10 mM NaCl, 1% NP-40) supplemented with protease inhibitor cocktail (Roche). Samples were diluted 1:1 in ChIP Dilution Buffer (0.01% SDS, 1.1% Triton, 0.2 mM EDTA; 16.7 mM Tris pH 8.1, 167 mM NaCl) and pulse sonicated using a Bioruptor sonicator (Diagenode, U.K.) at high power for 30 min on ice (15 s on, 15 s off). Sonicated lysates were clarified by centrifugation at 400 g for 10 min and precleared with Protein A agarose beads (Millipore, 16-156). Samples were immunoprecipitated with primary antibodies or IgG controls and pulled down with Protein A agarose beads. Precipitates were washed sequentially in low salt buffer (0.1% SDS, 1% Triton, 2 mM EDTA, 20 mM Tris pH 8.1, 150 mM NaCl), high salt buffer (0.1% SDS, 1% Triton, 2 mM EDTA, 20 mM Tris pH 8.1, 500 mM NaCl), LiCl Buffer (1% Igepal, 1 mM EDTA, 10 mM Tris pH 8.1, 250 mM LiCl, 1% sodium deoxycholate) and finally twice in TE wash buffer (10 mM Tris pH 8.0, 1 mM EDTA) before being eluted from the beads in 240 μL of elution buffer (0.1 M NaHCO₃, 1% SDS). Complexes were reverse crosslinked in a heated shaker at 65 °C overnight, 1400 rpm, in the presence of 200 mM NaCl. Eluates were treated with Proteinase K (SIGMA) and RNaseA (SIGMA) before cleanup using MiniElute PCR Purification columns. Samples were analyzed on a LightCycler96 (Roche) using SYBR green qPCR mastermix (PCR Biosystems, UK) and % Input was calculated for each sample relative to their own input controls.

**SDS–PAGE and western blots**. The cells were lysed in RIPA buffer (20 mM Tris, pH 7.5, 2 mM EDTA, 150 mM NaCl, 1% NP40, and 1% sodium deoxycholate) supplemented with protease inhibitor cocktail tablets (Roche). 4× reducing buffer was added to samples before incubating at 95 °C for 5 min. Proteins were separated on a 10% polyacrylamide gel and transferred to PVDF membranes (Amersham). Membranes were blocked in PBST, 5% skimmed milk (Sigma), and proteins detected using specific primary and HRP-secondary antibodies. Protein bands were detected using a Pierce SuperSignal West Pico chemiluminescent substrate kit (Pierce) and images collected with a PXi Touch Imaging system (Syngene).

**HBV infected human chimeric mice and RNA-seq analysis**. FRG-NOD mice were housed and bred at the INSERM U1110 animal facility (regional agreement n° E-67-482-7) and fed 2-(2-nitro-4-trifluoromethylbenzoyl)-1,3-cyclohexanedione (NTBC) in their drinking water (16 mg/L). Six-week-old FRG-NOD mice received $1.5 \times 10^9$ pfu of an adenoviral vector encoding for the urokinase-like plasminogen activator, and treated with NTBC (8 mg/L). Forty eight hours later mice were intrasplenically transplanted with $10^6$ PHHs (Life technologies) and fed NTBC at 0.8 mg/L. During the following days NTBC dose was decreased every 2 days to 0.4 mg/L and 0.2 mg/L, and finally withdrawn. Efficient transplantation was assessed 8 weeks later by measuring human serum albumin levels by ELISA (Bethyl). The transplantation procedure was approved by the local Ethics committee and authorized by the French ministry of research and higher education (APA-FIS#4485-2016031115352125 v3).

Successfully transplanted mice were infected with $10^9$ HBV genome equivalents purified from concentrated supernatant of the HepG2.2.15 cell line (HBV genotype

D subtype ayw)[77]. After eight weeks of infection mice were treated twice a day by intraperitoneal injection of either vehicle (15% Kolliphor EL, Sigma) or 100 mg/kg SR9009 for 2 weeks. The experimental procedure was approved by the local Ethics committee and authorized by the French ministry of research and higher education (APAFIS#13872-2018050214497349 v1). HBV DNA viral load was determined weekly using the clinically approved Abbott Real Time HBV assay (Abbott). Human albumin levels were determined as described previously[78]. Vehicle and SR9009 treated mice were sacrificed and livers harvested for RNA isolation and RNA-sequencing at Novagene. RNA purity was assessed with a NanoDrop 2000 spectrophotometer (Thermo Fisher Scientific) and integrity determined using a 2100 Bioanalyzer Instrument (Agilent Technologies). Sequence adapters were removed and reads trimmed by Trim Galore v0.5.0[79]. The reads were mapped against the reference mouse genome (mm10/GRCm38) and reference human genome (hg38/GRCm38) using STAR v2.5.3[80]. Counts per gene were calculated using Rsubread v1.28.1[81]. Reads were analyzed by edgeR v3.30.0[82], normalized using TMM, counts per million calculated and differential expression analysis performed. Mouse genes were converted to human orthologs using biomaRt v2.44.0 [83] and the built-in ensembl datasets for human and mouse. Pathway analysis using the Human KEGG 2019 library was performed using enrichR v2.1[84].

**Analysis of public REV-ERB microarray data**. The publicly available gene expression microarray dataset (GSE34018[52]) was obtained using GEOquery v2.56.0[85] and mapped to genes using illuminaMousev2.db v1.26.0. Two replicates at each of six timepoints (zeitgeber time 0, 4, 8, 12, 16 and 20) were used. Differential expression analysis, controlling for time of day, was performed using edgeR v3.30.0[82] with a p-value <0.05 cut off. Genes were converted to their human orthologs using biomaRt v2.44.0[83] and the ensembl datasets used to compare human and mouse transcriptomes.

**Normal and HBV infected liver biopsies**. Liver biopsies from chronic HBV infected ($n = 20$) and non-infected subjects ($n = 8$) were collected and a portion of the biopsy immersed in RNAlater and stored at −80 °C. Tissue was collected together with anonymized clinical and demographic data, with local ethical committee approval (CE90/19) for the use of this archival material in this study. RNA was isolated and the integrity and concentration assessed by Tapestation 2200 (Agilent Technologies).

**Statistical analysis**. All experiments were repeated at least three times. All data are presented as mean values ± SEM. p-values were determined using the Mann–Whitney test (two group comparisons) or with the Kruskal–Wallis ANOVA (multi group comparisons) using PRISM version 8. In the figures $*p < 0.05$, $**p< 0.01$, $***p<0.001$, $****p<0.0001$, and n.s. denotes non-significant.

**Reporting summary**. Further information on research design is available in the Nature Research Reporting Summary linked to this article.

## Data availability
The RNA-seq data from SR9009 treated mice are deposited at NCBI (GEO GSE163285) and can be accessed via: https://www.ncbi.nlm.nih.gov/geo/query/acc.cgi?acc=GSE163285. The authors declare that all data supporting the findings of this study are available within the article and its Supplementary Information files or are available from the authors upon request. Source data are provided with this paper.

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

## Acknowledgements

We wish to thank: Prof U Protzer (TUM, Germany) for providing HepG2-pEpi cells and HepaRG cells; Dr B Grimaldi (University of Genoa, Italy) for Lenti-shRev-Erbα constructs; Dr X Qin (Anhui University, Hefei, China) for Cry1 promoter construct, Bmal1 and Clock expression plasmids; Prof S Urban (University of Heidelberg, Germany) for Myrcludex B; Nicolas Brignon for technical help with animal work; Dina Kremsdorf for providing HBV for mouse infection studies and Alvina Lai for discussions on this project. This work was funded by Wellcome award IA 200838/Z/16/Z (JAM); Wellcome Institutional Strategic Support Fund 0005025 (JAM); MRC project grant MR/R022011/1 (JAM); MRC program grant MR/P023576/1 (DR); Wellcome 107849/A/15/Z (DR); Japan Agency for Medical Research and Development (AMED) J-PRIDE, JP19fm0208019j0003 (K.W.), AMED Program on the Innovative Development and the Application of New Drugs for Hepatitis B, JP19fk0310114j0003 (K.W.), and AMED Program for Basic and Clinical Research on Hepatitis, JP19fk0210036j0002 (K.W.). ARC, Paris (TheraHCC2 IHUARC2019 to T.F.B.), the European Union (EU H2020-667273-HEPCAR to J.A.M. and T.F.B., ANRS; LabEx HepSys and Inserm Plan Cancer (T.F.B.).

## Author contributions

X.Z. designed and conducted experiments and co-wrote the MS; D.F. designed and conducted experiments; S.T. designed and conducted experiments; V.D. designed and conducted experiments; L.M. designed and conducted experiments; J.M.H. designed and conducted experiments; P.A.C.W. designed and conducted experiments; H.B. designed and conducted experiments, M.S. designed and conducted experiments; A.M. designed and conducted experiments; C.O.R. provided technical support; R.J.M. analyzed data; M.I. analyzed data; M.G. provided reagents; R.M. provided clinical material; M.P. provided clinical material; S.B. provided reagents; P.B. analyzed data and co-wrote the MS; D.R. provided advice and co-wrote the MS; K.W. provided advice and reagents; T.F.B. co-wrote the MS: J.A.M. designed study and co-wrote MS.

## Competing interests

The authors declare no competing interests.
