## [Peer Review File · Nature Communications]

Reviewer comments, first round

Reviewer #1 (Remarks to the Author):

This is an interesting study that implicates REVERBa in the circadian regulation of the HBV receptor NTCP, and also suggests a role for BMAL1 in directly activating HBV DNA transcription in vivo. As REVERBa regulates BMAL1, use of the REVERB agonist SR9009 may modulate BMAL1 activity with therapeutic potential. In general, the experiments support the authors' conclusions. There are, however, a few points that should be addressed:

1. Fig 1c – quantification of blots. These rhythms are not particularly convincing. Quantification of the blots would be worthwhile to substantiate this using biological replicates.
2. Fig 2a – Are SR9009-treated cells less abundant than in the control – i.e. is there any cellular toxicity? DAPI co-staining (as in Fig. 2d) would enable an assessment of this. Otherwise it is not clear whether this is an SR9009 effect, or some degree of cell death.
3. Fig. 3e – There seem to be four independent blot fragments. Why weren't these done together, i.e. the REVERBa and actin blots should have the WT and KO lysates run on the same gel? In addition, the blots should (ideally) be quantified with biological replicates. The same is true for Supp Fig 3b – they should be on the same gel/blot for proper comparison.

The authors should include in their discussion a note about specificity of SR9009 for REVERBa, in view of data from the Lazar lab (PMID: 31127047) concerning this molecule.

Reviewer #2 (Remarks to the Author):

The manuscript „Circadian control of hepatitis B virus replication“ by Zhuang and colleagues investigates a little explored area in HBV infection, revealing novel virus-host cell interactions and potential targets for antiviral therapy. This multi-disciplinary study is likely to be of interest to a broad audience; however, more experimental data and statistical analysis are needed to support current conclusions and increase the significance of this work.

Specifically:

Fig. 1

REV-ERBa and NTCP expression are inversely correlated. However, when comparing REV-ERBa expression (mRNA, panel b) and NTCP levels (western blot, panel c), this is not evident; for example, at 6 and 18 h post-cell synchronization, REV-ERBa expression is lowest and highest, respectively, while NTCP level appears the same. Since there might be a delay between RNA expression and the level of the corresponding protein, the gel in panel c should be completed with REV-ERBa protein levels (as in fig 1d, the insert gel). Actin levels appear to vary considerably throughout the experiment described in panel c; therefore, quantification of BMAL1, REV-ERBa and NTCP levels normalized to actin is necessary to understand the circadian expression of these proteins.

In fig 1d, efficient REV-ERBa silencing was confirmed by western blot. REV-ERBa levels should also be assessed when overexpressed (panel d), for a relevant correlation with NTCP and BMAL1 levels. The mechanism of NTCP repression by REV-ERBa remains elusive. The NTCP promoter contains RORE and E-box sites for REV-ERBa and BMAL1 binding, respectively (page 7, Discussion). A ChIP analysis, performed as for HBV in Fig. 4, would provide more mechanistic insights into this repression.

Fig. 2

MyrB was included as the positive control for inhibition of HBV (Fig. 2B) and HDV (Fig. 2D) entry. The compound appears significantly more efficient in the latter case. Since HBV and HDV use the

same receptors to enter the cells, how is this difference explained?

Although the data in Fig1 indicates a circadian NTCP expression and function, the results in Fig. 2 are not proof that HBV infection follows the same pattern. Moreover, HBV relies on a second receptor for efficient endocytosis, EGFR, which has not been investigated in this work.

Fig. 3

This figure is very problematic to interpret as bits and pieces of experimental data are missing. SR9009 treatment of HepaRG cells, post-HBV infection, clearly results in reduction of the viral pgRNA but also of the HBsAg (Fig. 3a). Since expression of the envelope proteins is controlled by other promoters than BCP it is surprising that only BCP was considered in the reporter assay (Fig. 3b). Expression of all HBV transcripts should be shown in SR9009-treated cells, by Northern blot, to understand how this agonist regulates viral transcription in infected cells. The same applies for the HBV replicating cells (HepG2-pEpi) (Fig. 3c). The levels of secreted HBsAg and viral particles should also be investigated in these cells.

Fig. 4

Bmal1 appears well expressed in cells of hepatic origin, even when not synchronized for expression of circadian genes (Supplementary Fig. 2, panel b). What is the explanation for the lack of endogenous BMAL1 detection in HepG2-NTCP cells (Fig. 4b)?

The significantly lower binding of the bona fide HBV transcription factor HNF4A, in vivo (Fig. 4d) , as compared to in vitro (Fig. 4c) is intriguing. Please provide statistical analysis of the in vivo experiments, to understand the relevance of this positive control.

Fig. 5

A known target of the BMAL1 activator is Per1 (also used as control in Fig. 4d). It is therefore intriguing that Per1 expression is increased in SR9009-treated cells, both human and mouse (Fig. 5b).

The results regarding HBsAg secretion following SR9009 treatment in vitro (Fig. 3a) and in vivo (Fig. 5f) are conflicting. Given the modest inhibition of the HBV pgRNA synthesis in treated chimeric mice (Fig. 5e) it is crucial to quantify the viral titers in these animals, to understand the effect of SR9009 treatment on HBV particles production in vivo and hence, its clinical relevance.

Minor comments:

There are many discrepancies between figures and figure legends:

-Fig. 2c: two SR9009 concentrations were considered, 10 and 20 μ M (figure), not one (legend)

-Fig 2c: one week HBV inoculation (figure), 6 days (legend)

-Fig 2d: how long was the treatment, 24 (figure) or 48 hrs (legend)? What is the SR9009 concentration used in this experiment?

-Fig 3a: again, two SR9009 concentrations were considered, 10 and 20 μ M (figure), not one (legend)

Reviewer #3 (Remarks to the Author):

This manuscript describes experiments examining the effect of circadian control genes on hepatitis B virus replication. The authors have developed an in vitro experimental system enabling these studies, and also examined the effects in vivo, in human liver chimeric mice. They have shown that activation of the circadian factor REV-ERB reduces HBV infection and replication. This is mediated through direct binding of the circadian factor BMAL1 to the HBV DNA.

The paper is very well written, and interpretation of the experiments is logical. Interactions between host factors and HBV that may influence various viral processes has not been comprehensively studied, especially with regard to the influence of circadian factors. Therefore, this research is important and will be of great interest for the field. There are a few issues and suggestions especially regarding the inclusion of further materials and methods to allow reproduction of the techniques which the authors need to address.

Specific comments:

Results:

A major issue with the manuscript is how the authors have identified the HBV basal core promoter region that is a major focus of this study. The authors used a luciferase expression plasmid referred to as containing the HBV basal core promoter (BCP). This plasmid actually includes 900 bp of HBV DNA, that contains the HBV Enhancer I/X promoter, the entire X gene, and the Enhancer II/ core promoter. It is not accurate to refer to this as a BCP expression construct. The inclusion of these additional viral factors may influence the results obtained, and this should be discussed. In addition, the authors examined the HBV genome for motifs corresponding to the binding site of BMAL1 (E-box motifs), and identified a number of these motifs as within the BCP region. However the E-boxes 3 and 4 are not positioned within the BCP but are in the Enhancer I/ X promoter region, and E-Box 5 (identified as the binding site) is upstream of the BCP, the core upstream regulatory sequence (CURS) and Enhancer II.

In the results section, the authors refer to the construction of a knock-out (KO) Rev-Erb- α hepatocyte cell line, however the methods section details construction of a BMAL1 gene KO? Page 4, line 37: "Collectively, these data show a role for REV-ERB in repressing HBV BCP activity, associated pgRNA levels and particle genesis." This statement is incorrect, the results do not include examination of any effect on virion particle genesis. This is also suggested in the Discussion (page 6, line 12) where the authors state that; "BMAL1 binds HBV DNA and positively regulates the basal core promoter, increasing pgRNA levels and secretion of viral particles." No examination of the effect on viral secretion is included in the experiments described.

Page 7, line 4: The authors discuss experiments analysing the 1kb human NTCP promoter to identify RORE and E-box binding motifs, however this work is not shown.

Page 7, line 13-18: The reference to the experiments shown in Supplementary Figure 8 should be included in the Results section, not the Discussion. Also, in this experiment, the authors noted a reduction in BMAL1 in HBV infected patients, which the results indicate is significantly different to non-infected controls, however the authors have dismissed this result as "minimal". This result needs further discussion.

The authors have concentrated the study on the effects of BMAL1, however they also identified a RORE element (binding site for REV ERB) at the end of the PreS promoter. I understand that examining this is probably an additional separate study, however it would be interesting to include discussion of the potential for this to also influence HBV replication levels.

Methods:

Was the HDV inoculum obtained from cell culture or patient material, and how was it prepared?

The authors need to include a description of the NTCP promoter luciferase activity reporter plasmid. The authors have also used a Cry1 promoter construct, and BMAL/Clock expression plasmids. These also need describing.

The Bmal1 mRNA qPCR protocol needs to be included.

Can the authors please better explain the qPCR primers included in the table? What were the "HBV DNA update" and the "PrP" primers and probes used for? Which primers were used for the pgRNA PCR?

For the human liver chimeric mice, the authors comment that human albumin levels were determined as described previously, but have not included a reference for this.

In Figure 2 and Figure 3, the legends describe only the use of 20uM SR9009, however the graphs also show experiments using 10uM SR9009.

Figure 3 legend, typo "lysted" should be "lysed"?

There is no reference within the text for Figure 6?

Margaret Littlejohn

RE: Revised MS NCOMMS-20-222873R1 response to reviewers comments

Please find our point-by-point response to the reviewers comments

Reviewer #1. This is an interesting study that implicates REVERBa in the circadian regulation of the HBV receptor NTCP, and also suggests a role for BMAL1 in directly activating HBV DNA transcription in vivo. As REVERBa regulates BMAL1, use of the REVERB agonist SR9009 may modulate BMAL1 activity with therapeutic potential. In general, the experiments support the authors' conclusions. There are, however, a few points that should be addressed:

1. Fig 1c: quantification of blots. These rhythms are not particularly convincing. Quantification of the blots would be worthwhile to substantiate this using biological replicates. Response: We provide quantification of the western blots with biological replicates in new Fig.1c in our revised MS.

2. Fig 2a: Are SR9009-treated cells less abundant than in the control – i.e. is there any cellular toxicity? DAPI co-staining (as in Fig. 2d) would enable an assessment of this. Otherwise it is not clear whether this is an SR9009 effect, or some degree of cell death. Response: We provided new images that include DAPI staining in Fig.3a of our revised MS. To further address the effect of SR9009 on cell viability we include a new Supplementary Fig.2 that shows an LDH assay, confirming the limited cytotoxic effects of the agonist.

3. Fig. 3e: There seem to be four independent blot fragments. Why weren't these done together, i.e. the REVERBa and actin blots should have the WT and KO lysates run on the same gel? In addition, the blots should (ideally) be quantified with biological replicates. The same is true for Supp Fig 3b – they should be on the same gel/blot for proper comparison. Response: We apologise for this oversight and provide new images with all samples analysed on a single gel (current Fig.4e and Supplementary Figures 3 and 4). Since there are no bands to quantify in the Rev-erb α KO we consider it sufficient to just provide new images.

The authors should include in their discussion a note about specificity of SR9009 for REVERBa, in view of data from the Lazar lab (PMID: 31127047) concerning this molecule. Response: We have cited this paper in the discussion of our revised MS.

Reviewer #2: The manuscript "Circadian control of hepatitis B virus replication" by Zhuang and colleagues investigates a little explored area in HBV infection, revealing novel virus-host cell interactions and potential targets for antiviral therapy. This multi-disciplinary study is likely to be of interest to a broad audience; however, more experimental data and statistical analysis are needed to support current conclusions and increase the significance of this work. Specifically:

Fig. 1. REV-ERBa and NTCP expression are inversely correlated. However, when comparing REV-ERBa expression (mRNA, panel b) and NTCP levels (western blot, panel c), this is not evident; for example, at 6 and 18 h post-cell synchronization, REV-ERBa expression is lowest and highest, respectively, while NTCP level appears the same. Since there might be a delay between RNA expression and the level of the corresponding protein, the gel in panel c should be completed with REV-ERBa protein levels (as in fig 1d, the insert gel). Actin levels appear to vary considerably throughout the experiment described in panel c; therefore, quantification of BMAL1, REV-ERBa and NTCP levels normalized to actin is necessary to understand the circadian expression of these proteins. Response: We provide new experimental data showing REV-ERBa expression and quantification of the western blots demonstrate an inverse pattern of BMAL1 and NTCP (Fig.1c).

In Fig.1d, efficient REV-ERBa silencing was confirmed by western blot. REV-ERBa levels should also be assessed when overexpressed (panel d), for a relevant correlation with NTCP and BMAL1 levels. Response: We provide new western blot data showing Halo-tagged REV-ERBa expression (Fig.1e).

The mechanism of NTCP repression by REV-ERB α remains elusive. The NTCP promoter contains RORE and E-box sites for REV-ERB α and BMAL1 binding, respectively (page 7, Discussion). A ChIP analysis, performed as for HBV in Fig. 4, would provide more mechanistic insights into this repression. **Response:** We thank the reviewer for this suggestion and our revised MS contains new ChIP data that shows direct binding of REV-ERB α to the NTCP promoter (new Fig.2d-e).

Fig.2. MyrB was included as the positive control for inhibition of HBV (Fig.2B) and HDV (Fig. 2D) entry. The compound appears significantly more efficient in the latter case. Since HBV and HDV use the same receptors to enter the cells, how is this difference explained? **Response:** To assess the anti-viral effect of MyrB on HBV uptake into cells we used a recently reported assay to measure internalized virus particles by PCR quantification of trypsin-resistant DNA after 16h infection (Chakraborty *et al* 2020 Cell Microbiol). This assay uses a multiplicity of infection (MOI) of 200. In contrast, our HDV infection protocol uses an MOI of 25 and quantifies HDAg and RNA after 7 days. The differential blocking by MyrB may reflect the experimental models used and their respective viral doses.

Although the data in Fig1 indicates a circadian NTCP expression and function, the results in Fig. 2 are not proof that HBV infection follows the same pattern. Moreover, HBV relies on a second receptor for efficient endocytosis, EGFR, which has not been investigated in this work. **Response:** The extended inoculation times (24h) required to establish HBV infection limited our ability to infect synchronized dHepaRG cells at defined circadian times. We provide new data showing that SR9009 treatment of HepaRG cells had no effect on EGFR mRNA levels in Supplementary Fig.5.

Fig.3. This figure is very problematic to interpret as bits and pieces of experimental data are missing. SR9009 treatment of HepaRG cells, post-HBV infection, clearly results in reduction of the viral pgRNA but also of the HBsAg (Fig.3a). Since expression of the envelope proteins is controlled by other promoters than BCP it is surprising that only BCP was considered in the reporter assay (Fig. 3b). Expression of all HBV transcripts should be shown in SR9009-treated cells, by Northern blot, to understand how this agonist regulates viral transcription in infected cells. The same applies for the HBV replicating cells (HepG2-pEpi) (Fig.3c). The levels of secreted HBsAg and viral particles should also be investigated in these cells. **Response:** We really appreciate the reviewer's constructive comments. We now provide a northern blot of SR9009 treated HBV infected dHepaRG cells and show that the agonist reduces both pC/pgRNA and preS1/S2 transcripts (new Fig.4b), consistent with broad-reaching regulation of HBV promoters. This conclusion is supported by a reduction in HBeAg and HBsAg levels. Although the level of viral transcripts in the HepG2-pEpi cells was too low to obtain a publication quality northern blot we provide new data showing that SR9009 treatment significantly reduced the secretion of HBsAg and DNA containing particles (Fig.4d).

Fig.4. Bmal1 appears well expressed in cells of hepatic origin, even when not synchronized for expression of circadian genes (Supplementary Fig. 2, panel b). What is the explanation for the lack of endogenous BMAL1 detection in HepG2-NTCP cells (Fig. 4b)? **Response:** We apologise for omitting to state in Fig.4b (revised MS Fig.5b) legend that cells were transfected with a FLAG-tagged Bmal1 expression plasmid and blots were probed with anti-Flag to assess exogenous Bmal1 expression. We have clarified the figure legend in our revised MS.

The significantly lower binding of the bona fide HBV transcription factor HNF4A, *in vivo* (Fig.4d) , as compared to *in vitro* (Fig.4c) is intriguing. Please provide statistical analysis of the *in vivo* experiments, to understand the relevance of this positive control. **Response:** We appreciate the reviewer's comment and indeed the published literature studying transcription factor binding to HBV *in vivo* is limited. To consolidate our *in vivo* results, we provide further experimental ChIP data with chromatin isolated from three more liver tissues of the HBV infected mice and provide detailed statistical analysis. These new data clearly show Bmal1 binding to HBV genomes and the differences noted in HNF4a binding to viral genomes *in vitro* and *in vivo* may reflect differences in the HBV epigenome that we are currently investigating.

Fig.5. A known target of the BMAL1 activator is Per1 (also used as control in Fig.4d). It is therefore intriguing that Per1 expression is increased in SR9009-treated cells, both human and mouse (Fig.5b). **Response:** Our data showing that SR9009 treatment increased Per1/2 transcripts is consistent with a previous report (Solt *et al*, Nature, 2012). Further mechanistic experiments to explore this in more detail are beyond the scope of the current study.

The results regarding HBsAg secretion following SR9009 treatment *in vitro* (Fig. 3a) and *in vivo* (Fig. 5f) are conflicting. **Response:** Previous studies have reported differences in the level of HBsAg secreted from HBV infected human hepatoma cell lines (Li *et al* 2016 J Virol and Sozzi *et al* 2016 J Virol), however, no mechanism was provided. Our new northern blot data (revised MS Fig.4b) shows that treating infected dHepaRG cells with SR9009 reduced pre-S1/S2 RNA levels, demonstrating a role for the agonist to repress HBsAg at the transcriptional level. It is likely that epigenetic differences in the HBV genome exist *in vitro* and *in vivo* and this may regulate different promoter usage and is an understudied area of HBV biology.

Given the modest inhibition of the HBV pgRNA synthesis in treated chimeric mice (Fig. 5e) it is crucial to quantify the viral titers in these animals, to understand the effect of SR9009 treatment on HBV particles production in vivo and hence, its clinical relevance. Response: Dusseaux et al 2017 Gastroenterology reported on the dynamics of peripheral HBV DNA loss in infected liver chimeric mice treated with Entecavir, that directly targets reverse transcription conversion of pgRNA-rcDNA. Even after 4 weeks of treatment there was only partial suppression with significant variability observed between animals (shown in Fig.7 and Supplementary Fig.9a). We have included HBV DNA levels in the revised MS (Fig.6f). Given the short duration of treatment (2 weeks), it is unsurprising that peripheral HBV DNA levels were not affected.

Minor comments: There are many discrepancies between figures and figure legends:

Fig.2c: two SR9009 concentrations were considered, 10 and 20 μ M (figure), not one (legend). Response: Apologies we have corrected the figure legend in the revised MS.

Fig 2c: one week HBV inoculation (figure), 6 days (legend). Response: Apologies we have corrected the figure legend in the revised MS.

Fig.2d: how long was the treatment, 24h (figure) or 48h (legend)? What is the SR9009 concentration used in this experiment? Response: Apologies we have corrected the figure legend to correctly state the duration (48h) and dose of SR9009 (20 μ M) used for these experiments.

Fig3a: again, two SR9009 concentrations were considered, 10 and 20 μ M (figure), not one (legend). Response: Revised legend states dose of SR9009 used in all experiments.

Reviewer #3: This manuscript describes experiments examining the effect of circadian control genes on hepatitis B virus replication. The authors have developed an in vitro experimental system enabling these studies, and also examined the effects in vivo, in human liver chimeric mice. They have shown that activation of the circadian factor REV-ERB reduces HBV infection and replication. This is mediated through direct binding of the circadian factor BMAL1 to the HBV DNA. The paper is very well written, and interpretation of the experiments is logical. Interactions between host factors and HBV that may influence various viral processes has not been comprehensively studied, especially with regard to the influence of circadian factors. Therefore, this research is important and will be of great interest for the field. There are a few issues and suggestions especially regarding the inclusion of further materials and methods to allow reproduction of the techniques which the authors need to address.

Specific comments: A major issue with the manuscript is how the authors have identified the HBV basal core promoter region that is a major focus of this study. The authors used a luciferase expression plasmid referred to as containing the HBV basal core promoter (BCP). This plasmid actually includes 900 bp of HBV DNA, that contains the HBV Enhancer I/X promoter, the entire X gene, and the Enhancer II/ core promoter. It is not accurate to refer to this as a BCP expression construct. The inclusion of these additional viral factors may influence the results obtained, and this should be discussed. Response: We apologise for this omission and have provided a more detailed description of the promoter construct used in our studies in the revised MS.

In addition, the authors examined the HBV genome for motifs corresponding to the binding site of BMAL1 (E-box motifs) and identified a number of these motifs as within the BCP region. However the E-boxes 3 and 4 are not positioned within the BCP but are in the Enhancer I/ X promoter region, and E-Box 5 (identified as the binding site) is upstream of the BCP, the core upstream regulatory sequence (CURS) and Enhancer II. Response: We apologise for this omission and we have expanded the text accordingly.

In the results section, the authors refer to the construction of a knock-out (KO) Rev-Erb- α hepatocyte cell line, however the methods section details construction of a BMAL1 gene KO? Response: We apologise for this typographical error and we confirm that our experiments used a Rev-ERB KO line and have corrected the Methods section.

Page 4, line 37: "Collectively, these data show a role for REV-ERB in repressing HBV BCP activity, associated pgRNA levels and particle genesis." This statement is incorrect, the results do not include examination of any effect on virion particle genesis. This is also suggested in the Discussion (page 6, line 12) where the authors state that; "BMAL1 binds HBV DNA and positively regulates the basal core promoter, increasing pgRNA levels and secretion of viral particles." No examination of the effect on viral secretion is included in the experiments described.

Response: Our original MS showed that SR9009 treatment of HepG2.2.15 cells reduced extracellular HBV DNA (Fig.3d – now Supplementary Fig.6 in MS R1), supporting our statement on page 4, line 37. To support this conclusion and to complement our data set with HepG2-pEpi cells we present a new figure showing that SR9009 treatment reduces extracellular DNA and HBsAg (Fig.4d).

Page 7, line 4: The authors discuss experiments analysing the 1kb human NTCP promoter to identify RORE and E-box binding motifs, however this work is not shown. Response: We now show RORE and E-box motifs in the -1kb human NTCP promoter (Fig.2d-e). We also provide a new ChIP analysis showing an enrichment of REV-ERB α binding to the endogenous NTCP promoter (Fig.2d), providing direct evidence for REV-ERB to act as a repressor of NTCP.

Page 7, line 13-18: The reference to the experiments shown in Supplementary Figure 8 should be included in the Results section, not the Discussion. Also, in this experiment, the authors noted a reduction in BMAL1 in HBV infected patients, which the results indicate is significantly different to non-infected controls, however the authors have dismissed this result as “minimal”. This result needs further discussion. Response: We agree that the change in BMAL1 and REV-ERB transcripts observed in liver tissue from patients with chronic hepatitis B compared to normal healthy controls is interesting. Since all of the patients are from a single Italian cohort we are keen to extend these observations and to compare HBeAg positive and negative patients with different viral loads. We are hoping this will form the foundation of an independent report.

The authors have concentrated the study on the effects of BMAL1, however they also identified a RORE element (binding site for REV-ERB) at the end of the PreS promoter. I understand that examining this is probably an additional separate study, however it would be interesting to include discussion of the potential for this to also influence HBV replication levels. Response: REV-ERB is notoriously difficult to ChIP, however, encouraged by our results with the NTCP promoter we assessed REV-ERB binding to HBV DNA isolated from HepG2-pEpi cells (Supplementary Fig.9). We failed to show any evidence of REV-ERB binding to HBV despite showing excellent enrichment with the host target Bmal1.

Methods:

Was the HDV inoculum obtained from cell culture or patient material, and how was it prepared? Response: We provide information on the HDV infection protocol in the revised MS.

The authors need to include a description of the NTCP promoter luciferase activity reporter plasmid. The authors have also used a Cry1 promoter construct, and BMAL/Clock expression plasmids. These also need describing. The Bmal1 mRNA qPCR protocol needs to be included. Response: We have provided this information in the methods section of our revised MS.

Can the authors please better explain the qPCR primers included in the table? What were the “HBV DNA update” and the “PrP” primers and probes used for? Response: We apologise this was an error and was an internal note to identify that these primers were used to measure HBV internalization, this has been corrected in the revised MS.

Which primers were used for the pgRNA PCR? Response: For HBV pgRNA, we used the Taqman probe purchased from ThermoFisher. The information is listed in the Oligonucleotides table in the Methods section of our revised MS.

For the human liver chimeric mice, the authors comment that human albumin levels were determined as described previously, but have not included a reference for this. Response: We have provided the citation (Mailly et al. Nature Biotechnology 2015) in our revised MS.

In Figure 2 and Figure 3, the legends describe only the use of 20uM SR9009, however the graphs also show experiments using 10uM SR9009. Response: We apologise for this error and the legends for these figures are now correct in the revised MS.

Figure 3 legend, typo “lysted” should be “lysed”? Response: This typographical error is corrected in the revised MS.

There is no reference within the text for Figure 6? Response: Now cited in discussion of the revised MS

Reviewer comments, second round

Reviewer #1 (Remarks to the Author):

The authors have significantly revised their manuscript to address all of the issues raised by me and the other reviewers. I have no further issues.

Reviewer #2 (Remarks to the Author):

The authors have answered most of this reviewer's concerns. The manuscript has gained clarity, the major claims are well justified experimentally within the limits of the model systems currently available to study HBV infection. The work is original and provides novel, relevant and much needed information on the HBV life-cycle, that will certainly be of interest to researchers in the HBV field and beyond.

Reviewer #3 (Remarks to the Author):

Thank you for addressing all my suggestions and comments regarding the manuscript satisfactorily. I am happy with the response.

Margaret Littlejohn